# Acetone Gas Sensor Based on SWCNT/Polypyrrole/Phenyllactic Acid Nanocomposite with High Sensitivity and Humidity Stability

**DOI:** 10.3390/bios12050354

**Published:** 2022-05-19

**Authors:** Jun-Ho Byeon, Ji-Sun Kim, Hyo-Kyung Kang, Sungmin Kang, Jin-Yeol Kim

**Affiliations:** 1School of Advanced Materials Engineering, Kookmin University, Seoul 136-702, Korea; qus0414@naver.com (J.-H.B.); yyjbbj90@kookmin.ac.kr (J.-S.K.); rkdgyrud7274@kookmin.ac.kr (H.-K.K.); 2Head Office Laboratory, LG Japan Lab Inc., Tokyo 140-0002, Japan; sungmin1.kang@lgjlab.com; 3Institute of Innovative Research Laboratory for Future Interdisciplinary Research, Tokyo Institute of Technology, Yokohama 226-8503, Japan

**Keywords:** acetone gas sensor, polypyrrole-SWCNT composite, high sensitivity, room temperature, moisture resistance

## Abstract

We synthesized core-shell-shaped nanocomposites composed of a single-walled carbon nanotube (SWCNT) and heptadecafluorooctanesulfonic acid-doped polypyrrole (C8F-doped-PPy)/phenyllatic acid (PLA), i.e., C8F-doped-PPy/PLA@SWCNT, for detecting acetone gas with high sensitivity and humidity stability. The obtained nanocomposites have the structural features of a sensing material as a C8F-doped-PPy layer surrounding a single-stranded SWCNT, and a PLA layer on the outer surface of the PPy as a specific sensing layer for acetone. PLA was chemically combined with the positively charged PPy backbone and provided the ability to reliably detect acetone gas at concentrations as low as 50 ppb even at 25 °C, which is required for medical diagnoses via human breath analysis. When C8F was contained in the pyrrole monomer in a ratio of 0.1 mol, it was able to stably detect an effective signal in a relative humidity (RH) of 0–80% range.

## 1. Introduction

Noninvasive and portable breath analysis sensors are of increasing interest for monitoring a variety of diseases, including diabetes [1,2,3,4,5]. The breath that a normal person exhale contains approximately 5% water (H_2_O) vapor and carbon dioxide (CO_2_); approximately 1 ppm ammonia (NH_3_); and ~1 ppm acetone (C_3_H_6_O), methanol (CH_3_OH), ethanol, and other volatile organic compounds. Among these exhaled gases, the concentration of C_3_H_6_O is related to metabolism. Moreover, it has been reported that a strong association exists between the concentration of C_3_H_6_O in the breath and the rate of body fat loss [6,7,8,9]. Additionally, breath C_3_H_6_O is regarded as a biomarker for diabetes [9,10,11]. In general, the concentration of breath C_3_H_6_O in healthy individuals is below 1 ppm, but in diabetic patients, this can be twice as much. However, there are still major challenges for disease-detection sensors using respiratory gases, as follows: (1) the responsiveness of ppb levels is required because of the low concentration of the gases in the breath; (2) high selectivity is needed because of the complex composition of the breath, which includes a large quantity of moisture; and (3) for practical applications, such as point of care, the sensor must be instantaneous, easy to use, and portable and must function at a low operating temperature.

Semiconductor metal oxide (SMO)-based gas sensors have been reported as one of the most promising devices for practical sensing due to their major advantages of high sensitivity and fast response and recovery times [12,13]. In particular, SMO materials such as zinc oxide [14], indium (III) oxide [15], tin (IV) oxide [16], and tungsten trioxide [17] are being developed for C_3_H_6_O gas detection. However, technical problems remain in the practical application of this type of SMO sensor. First, current-sensing materials can hardly meet the response requirements for levels of tens of ppb to trace the C_3_H_6_O gas released from respiration. Second, the large amount of moisture that exists in exhaled breath severely interferes with C_3_H_6_O detection. As a result, the detection of low concentrations of C_3_H_6_O under high-humidity conditions is extremely limited. Third, since SMO-based sensors generally operate at a high temperature of 200–400 °C, an additional device for controlling the temperature of the sensor substrate is required.

To solve these problems, we synthesized core–shell-shaped carbon nanocomposites comprising a single-walled carbon nanotube (SWCNT) core and heptadecafluorooctanesulfonic acid-doped polypyrrole (C8F-doped-PPy)/phenyllatic acid (PLA) shells, i.e., C8F-doped-PPy/PLA@SWCNT, for C_3_H_6_O gas sensing at a sub-ppm level at room temperature and in stable humidity (see Figure 1). Structurally, it is a rod-shaped conductive carbon nanocomposite in which single-strand SWCNTs are embedded in C8F-doped-PPy, and the surface of C8F-doped-PPy is surrounded by PLA molecules that selectively react to C_3_H_6_O gas. Specifically, C8F is arranged on the surface of conductive PPy and acts as a dopant that not only affects the electrical resistance value, but also minimizes the changes in resistance due to moisture because of its hydrophobic properties. Recently, these chemical sensors based on π-conjugated conductive carbons (CCCs), such as carbon nanotubes (CNTs) [18,19,20], PPy [21,22,23], polyaniline [24], and graphene [25], are receiving a lot of attention as a gas-sensing material due to their changes in resistance when reducing or oxidizing gases are absorbed. The electrical resistance of these CCCs affects conductivity as the transfer of charged carriers within the π-conjugated chain and their resistance is changed by the adsorption of gas species. That is, when the CCC-based sensor is exposed to a gas such as C_3_H_6_O, their resistance is reduced in proportion to the concentration of the adsorbed gas. These organic-based C_3_H_6_O sensors have two advantages: they always respond at room temperature and can be easily designed as a material with high selectivity and sensitivity. Recently, Chung et al. [2] reported a C_3_H_6_O sensor based on cylindrical nanopore carbon composite with enhanced responsiveness down to the ppb level.

In this study, we synthesized C8F-doped-PPy/PLA@SWCNT with a diameter of 40 nm and length of 1–5 μm via emulsion polymerization. The obtained nanorcomposites have the structural features of a sensing material as a C8F-doped-PPy layer surrounding a single-stranded SWCNT, and a PLA layer on the outer surface of the PPy as a specific sensing layer for acetone. These sensors provided the ability to reliably detect acetone gas at concentrations as low as 50 ppb even at room temperature, and was able to stably detect an effective signal in a relative humidity (RH) in the range of 10–80%. This is an essential item for medical diagnoses through analysis of human breathing gas.

## 2. Materials and Methods

### 2.1. Synthesis of C8F-Doped-PPy/PLA@SWCNT

C8F-doped-PPy/PLA@SWCNT nanocomposites were synthesized via oxidative chemical polymerization according to a simple one-step emulsion polymerization method from a previous report [21]. First, hydroxypropylmethyl cellulose (HPMC, Sigma-Aldrich, St. Louis, MO, USA) as dissolved in deionized (DI) water to prepare a 1 wt% aqueous solution. Then, 0.0031 g of SWCNTs (Zeon, Japan) was placed in 30 mL of 1 wt% HPMC solution and dispersed through sonication for 30 min. Following this, 0.3 g of pyrrole monomer was added, and the mixture was stirred for 30 min to create solution A. Second, 0.37 g of L-(-)-3-phenyllactic acid (PLA) was dissolved in 10 mL of 1 wt% HPMC solution. The resulting solution was added to solution A, and the mixture was stirred for 30 min to create solution B. Third, 3.11 g of iron chloride hexahydrate (FeCl_3_∙6H_2_O) was dissolved in 30 mL of 1 wt% HPMC solution, and then the resulting solution was mixed with solution B and reacted at 4 °C for 6 h to proceed with the synthesis of solution C. Finally, 560 µL of 40 wt% aqueous C8F solution (pyrrole, C8F = 1:0.1 molar ratio) was added to solution C as a dopant, and the mixture was stirred at 300 rpm for more than 12 h. The synthesized core–shell-structured nanocomposite sensing materials were washed 2–3 times with CH_3_OH and DI water and finally prepared as a CH_3_OH dispersion ink solution.

### 2.2. Structural Analysis and Measurement of Gas Sensors

As in the normal process, a silicon substrate with a pair of Au electrodes (test cells) printed on the top surface was used. The as-synthesized solution inks for C8F-doped-PPy/PLA@SWCNT were coated onto a sensor substrate to form sensing layers via dropping and were subsequently dried at 80 °C for 10 min. The prepared gas sensor was installed in the test cell and the resistance value change according to the gas injection was measured using an electrochemical detection technique. To observe the gas-detection performance of the sensing layer, a test cell was placed in a 350 mL gas chamber. Pure dry air was injected as the carrier gas for 10 min while maintaining the chamber pressure at 100 torr. Then, C_3_H_6_O gas was exposed to various concentrations in the range of 0.05–5 ppm at 25 °C in dry air (0% relative humidity (RH)) and under wet conditions (10–80% RH). The resistances between electrodes were measured under a constant DC applied current. The current resistance (Ω, ohm) was measured using a resister (Keithley 2000) by applying a DC voltage of 2 V.

## 3. Results

A schematic structure of the C8F-doped-PPy/PLA@SWCNT core–shell-shaped nanocomposites for C_3_H_6_O gas sensing is shown in Figure 1. Figure 2 shows the morphological and structural characteristics of C8F-doped-PPy/PLA@ SWCNT nanocomposite investigated by SEM and TEM. Figure 2I displays a SEM image of the C8F-doped-PPy/PLA@SWCNT nanocomposite, showing well-defined rod-like morphology with lengths of 1–5 μm and an average diameter of approximately 40 nm. The TEM images of the regions that are specific parts of the SEM are presented in Figure 2II. As a result of TEM analysis, it was found that SWCNTs have a typical core-shell structure impregnated with C8F-doped-PPy and were in the form of single coaxial nanorods. The outer surface of a single strand of SWCNT with 2 nm is surrended by a layer of PPy with a thickness of about 15–17 nm, and a thin organic PLA layer of 2.5 nm covered the surface layer of PPy. Figure 2II shows the surface plasmon absorption spectrum of the sample. The bandgap peak associated with the π–π* transition of the chain backbone of PPy and the peak attributed to the positively charged polaron transport present in the conjugated chain were measured at 472 and 1060 nm, respectively. The electrical resistance (Ω) of the sensor sample of the as-cast film that formed as a function of the relative thickness (thickness was converted concerning the light transmittance (%) at a 550 nm wavelength of the casting film) is also shown in Figure 2IV. As plotted in Figure 2IV, the sensor films showed resistance values of 3, 4, 5, 7, 13, 18, and 24 kΩ at a relative thickness with transmittance values of 25%, 35%, 45%, 50%, 60%, 65%, and 75%, respectively.

Figure 3Ia shows the FT-IR spectra of the C8F-doped-PPy/PLA@SWCNT nanocomposite. For the structural analysis of the composite, Figure 3Ib,c also shows the IR spectra of the pure PLA sensing molecules in the outer shell and pure C8F-doped-PPy@SWCNT in the core, respectively. As shown in Figure 3Ia, all peaks of the C8F-doped-PPy/PLA@SWCNT show nearly identical wavenumbers and positions for the main IR bands, which are attributed to the structure of the SWCNT@C8F-doped-PPy and PLA. However, the IR spectrum data of SWCNTs present in a small amount of 0.008 wt% were not observed because the absorption was too weak. According to existing studies [23,26,27], the characteristic peaks at 1655, 1552, and 1470 cm^−1^ can be associated with C–C and C–N symmetric and asymmetric stretching; the peaks at 1300 and 1182 cm^−1^ can be attributed to C–H and C–N in-plane deformation vibrations, respectively; and the peaks near 1050 and 923 cm^−1^ can be attributed to the C–H deformation and N–H stretching vibrations. A specific absorption band, which resulted because of charge–charge coupling between the negatively charged COO^−^ in PLA molecules and the positive charge (arising C^+^ from positive polaron) of the PPY backbone, was observed at the 1400 cm^−1^ position, as shown in Figure 3Ia [23]. From these IR data, we confirmed that functional PLA molecules, which selectively reacted with C_3_H_6_O gas in the positively charged PPy chain on the surface of the C8F-doped-PPy@SWCNT conductive carbon nanocomposite, were strongly coupled (see the molecular schematic of Figure 1).

Figure 3IIb,c shows the Raman spectra of the pure C8F-doped-PPy and pure SWCNT, respectively, for the component analysis of the composite. In Figure 3IIc, the 1344 and 1590 cm^−1^ peaks correspond to the D band and G band (tangential mode related to the E2g symmetry) shown in the SWCNT backbone, respectively, and the 1320 and 1560 cm^−1^ peaks in Figure 3II,b can also be assigned as C–H/C–N and C=C stretch bands due to the quinoid structure of PPy according to the previous reports [28,29,30,31]. As a result, we can see that the 1340 and 1560 cm^−1^ peaks in Figure 3II,a are marked by overlapping C-H/C-N and C=C stretching modes of the PPy and D and G bands of SWCNTs, respectively. However, the intensity of the Raman absorption peak of C8F-doped-PPy/PLA@SWCNT was also significantly increased because of the stretching mode of C8F-doped-PPy and tangential mode of SWCNT overlapping. However, using Raman spectra, we were able to confirm the existence of SWCNTs presented in the molecular schematic in Figure 1 and the TEM image in Figure 2.

A synthetic ink solution comprising C8F-doped-PPy/PLA@SWCNT as a sensing layer for a sensor array and C_3_H_6_O gas-response measurement was drop-coated onto a sensor. The surface morphology of the sensor film comprised one-dimensional C8F-doped-PPy/PLA@SWCNT core–shell nanorods with a microporous structure that appeared to be intertwined within a three-dimensional network, as shown in the SEM morphology of Figure 2I. These porous surfaces could also be characterized by increasing the surface area that was in contact with the gas. Figure 4 shows representative data illustrating the response characteristics of the sensor array in response to C_3_H_6_O.

Figure 4Ia shows the continuous dynamic response of the C8F-doped-PPy@SWCNT (PLA-free) sensor to various concentrations of C_3_H_6_O (1, 2.5, and 5 ppm) at 25 °C and in 0% RH (dry air). Here, the response showed a decrease in the sensor response (ΔR/Ri × 100, S%) upon being exposed to various concentrations of C_3_H_6_O gas, where ΔR = R − Ri, Ri is the initial current resistance, and R the changed current resistance after exposure to C_3_H_6_O gas. The sensitivity (ppm^−1^) of a gas sensor is defined as the slope of the response, S, to the C_3_H_6_O gas concentration. As shown in Figure 4Ia, the response characteristics of C8F-doped-PPy@SWCNT (PLA-free) were indicated as 0.02, 0.04, and 0.09 at each C_3_H_6_O concentration (1, 2.5, and 5 ppm, respectively), and as the concentration increased, the relative response also increased, but their values were not significantly high. According to existing studies [21,32,33], PPy or SWCNT/PPy-composited compounds generally exhibit a similar change in electrical resistance according to the typical characteristics of a p-type semiconductor. We also reported on ammonia and C_3_H_6_O gas detection sensors based on nanocomposites of SWCNTs and PPy in a previous study [21]. Ammonia gas showed a high detection performance of 100 ppb, but acetone gas could not effectively detect a low concentration of less than 1 ppm. The decrease in electrical resistance of the C8F-doped-PPy@SWCNT sensor, as a p-type semiconductor, when exposed to C_3_H_6_O gas was attributed to the charge–charge interaction between C_3_H_6_O molecules acting as an acid and the positively charged polaron of the PPy backbone (see Figure 4I). As a result, upon the interaction of C_3_H_6_O with PPy, the PPy received its electron from C_3_H_6_O, and this electron transfer between PPy positive charges and C_3_H_6_O caused an elevation in charge-carrier concentration, which resulted in a decrease in the overall electrical resistance. Therefore, as shown in Figure 4I, as the concentration of C_3_H_6_O increases, the change in resistance of the sensor significantly increases in a negative direction. Figure 4Ib–d shows the continuous dynamic response of an C8F-doped-PPy/PLA@SWCNT (PLA-included) sensor to C_3_H_6_O concentrations. Here, PLA molecules could receive electrons from C_3_H_6_O when they were in contact with C_3_H_6_O molecules, which acted as acids; as the result of this, part of the hydroxyl group was converted to OH_2_^+^ to increase the positively charged density of the PPy backbone by self-doping. Experimentally, as shown in Figure 4Id, when PLA groups were linked in a 0.5 mol ratio to the positively charged PPy backbone, the response characteristics improved more than 10-fold compared to when the PLA group was absent, and the increase rate showed a change in proportion to the PLA concentration. However, in the C8F-doped-PPy/PLA@SWCNT sensor, PLA molecules were characterized as having an effective reactivity with the selectivity for C_3_H_6_O. The effect of increasing the concentration of C_3_H_6_O gas on the sensing parameters of C8F-doped-PPy/PLA_0.5_@SWCNT (PLA including a 0.5-mol ratio to PPy) was investigated with C_3_H_6_O in the 50–100 ppb concentration range at 25 °C and in dry air. The reversible and reproducible responses of the C8F-doped-PPy/PLA_0.5_@SWCNT with an injection of 50 and 100 ppb of C_3_H_6_O are shown in Figure 4II. As shown in Figure 4IIb, a significant signal was observed in trace gas of 50 ppb. On the other hand, in the C8F-doped-PPy/PLA_0.__0_@SWCNT (PLA-free) case, effective detection was not observed with the injection of 50 and 100 ppb of C_3_H_6_O. This result suggests that the reactivity is more activated by charge–charge coupling between the OH group of the PLA molecule linked to the PPy chain and the oxygen atom in the ketone group of the acetone molecule.

Figure 5a presents the plotting of the change in sensitivity of the C8F-doped-PPy/PLA_0.5_@SWCNT sensor when exposed to C_3_H_6_O gas at various concentrations ranging from 50 to 5000 ppb. As shown in the figure, the value of the linear correlation coefficient of the fitting curve for the sensitivity to gas concentration was obtained. However, the C8F-doped-PPy/PLA_0.5_@SWCNT sensor was shown to work linearly at concentrations as low as 50 ppb. Also, for comparison proposals, we compared changes in gas responsiveness of the C8F-doped-PPy/PLA_0.__0_@SWCNT sensor under the same conditions, as shown in Figure 5b. However, in the case of C8F-doped-PPy/PLA_0.__0_@SWCNT sensor, no significant data were obtained at concentrations below 1 ppm. The results indicate that the C8F-doped-PPy/PLA_0.5_@SWCNT sensor operates linearly down to concentrations as low as 50 ppb, showing higher sensitivity than C8F-doped-PPy/PLA_0.__0_@SWCNT sensor, as shown in the figure.

Moisture is an important influencing factor practically considered in the diagnosis of diseases such as diabetes, using respiratory gas. Therefore, obtaining reliable responses to C_3_H_6_O ppb levels and avoiding cross sensitivity due to large amounts of moisture in exhaled breath remain major challenges [13]. Almost all CCC-based sensors, including SMO sensors, are easily influenced or do not respond in high-RH conditions. In this study, a hydrophobic PPy nanocomposite (C8F-doped-PPy/PLA0.5@SWCNT) was synthesized using a perfluoroalkyl compound (C8F) as a dopant during the synthesis process to ensure the humidity stability of the sensor material. According to a previous paper [34], C8F not only serves as a dopant, but also can be used to make the PPy surface hydrophobic and is widely used to impart materials with water-repellent properties. In this work, we successfully synthesized core–shell-shaped C8F-doped-PPy/PLA_0.5_@SWCNT nanorods as a C_3_H_6_O-sensing material, containing C8F in a ratio of 0.1 mol to pyrrole monomer. As a result, it was confirmed that stable sensor detection was possible even under 80% RH humidity conditions. On the other hand, when the C8F content was contained in a large amount of 0.1 mol/ratio or more, humidity stability was improved, but the responsiveness of the sensor rapidly decreased at low temperature conditions. The effect of increasing the concentration of C_3_H_6_O on the continuous and dynamic response of the C8F_0.1_-doped-PPy/PLA_0.5_ @SWCNT sensor film was measured in the 1–5 ppm concentration range and under various humidity conditions.

Figure 6I shows the sensitivity of the C8F_0.1_-doped-PPy/PLA_0.5_@SWCNT sensor to C_3_H_6_O at a concentrations of 1 and 5 ppm under humidity conditions ranging from 0 to 80% RH at 25 °C. At this time, in the case of 5 ppm C_3_H_6_O gas, the observed response values were 1.45, 0.85, and 0.55, and 0.05 under the 0, 25, 50, and 80% RH conditions, respectively, as shown in the figure. The sensor response showed a tendency to decrease inversely proportional to the amount of humidity. The same experiment was also performed for C_3_H_6_O gas at 1 and 2.5 ppm, and a calibration curve was obtained by plotting sensor response as a function of humidity concentration. As shown in Figure 6II, when C_3_H_6_O gas was injected at 5 ppm, S decreased linearly, inversely proportional to humidity, and a significant value could be detected even under 80% RH conditions. However, S showed a tendency to decrease rapidly when the relative humidity was at least 10% or more, and in the case of a 1 ppm concentration, the response signal could hardly be measured under conditions of 50% RH or more. This result was believed to have occurred because the adsorption of a small amount of C_3_H_6_O gas to the surface of small droplets increased as the humidity increased. On analyzing the data in the figure, S of 5 ppm C_3_H_6_O under dry conditions was 1.5 but decreased to 0.75 at 40% RH, 0.4 at 60% RH, and 0.05 at 80% RH. This S value was approximately consistent with those of the 2.5, 1, and 0.5 ppm values of C_3_H_6_O measured under dry conditions. These results predicted that C_3_H_6_O gas corresponding to approximately 2.5 ppm at 40% RH, approximately 4 ppm at 60% RH, and nearly 4.5 ppm at 80% RH would be adsorbed by water droplets in a high-humidity environment.

According to the results of previous reports [35,36], when gaseous C_3_H_6_O molecules comes into contact with the water droplet, they can be easily absorbed into the water droplet due to the strong interaction between the carbonyl oxygen atom of C_3_H_6_O and the hydrogen atom of the OH group of the water molecule. It was also described as being more affected by low-temperature conditions. However, as a result of this experiment, it is thought that the decrease in the detection performance of acetone as the humidity increases is actually because a larger number of acetone gas is adsorbed onto the water droplets under high-humidity conditions, and as a result, the concentration of acetone gas is relatively reduced. For this reason, it is considered that an effective signal could not be detected under a humidity condition of 40% RH or higher at a concentration of 1 ppm acetone.

## 4. Conclusions

In this study, we synthesized C8F-doped-PPy/PLA@SWCNT nanocomposites with a diameter and length of 40 nm and 1–5 μm, respectively, via emulsion polymerization as C_3_H_6_O-sensing materials. Using SEM, TEM, and several spectroscopic analyses, the obtained sensing material reflected the structural features of the novel sensing material as a C8F-doped-PPy layer wrapped around a single-stranded SWCNT, with a PLA layer on the outer surface of PPy as a specific sensing layer. Here, the PLA bonded chemically with the PPy backbone, exhibited a sensing synergistic effect, and greatly improved the sensitivity to C_3_H_6_O. Finally, C8F-doped-PPy/PLA@SWCNT as a C_3_H_6_O sensor also provided reliable detection at a low concentration of 50 ppb at 25 °C. In addition, it was able to stably detect an effective signal in an RH range of 0–80% even at a low temperature, which is required for medical diagnoses via human breath analysis. However, in this study, unfortunately, we could not obtain a valid detection signal under the humidity condition of 40% RH or higher at an acetone concentration of 1 ppm.

## Figures and Tables

**Figure 1 biosensors-12-00354-f001:**
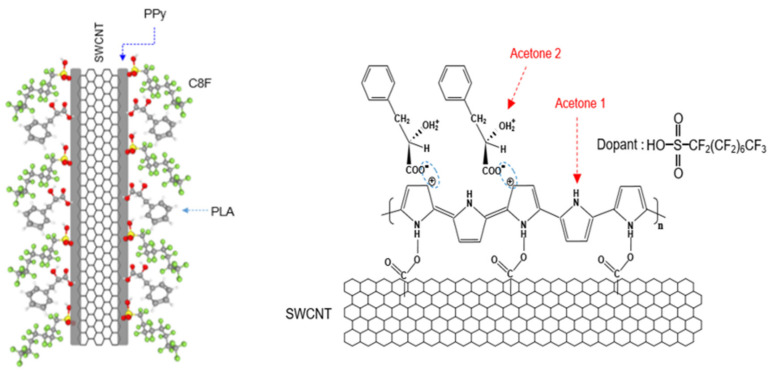
Schematic of and the chemical structure of C8F-doped-PPy/PLA@SWCNT core-shell-shaped nanocomposites for C_3_H_6_O gas sensing.

**Figure 2 biosensors-12-00354-f002:**
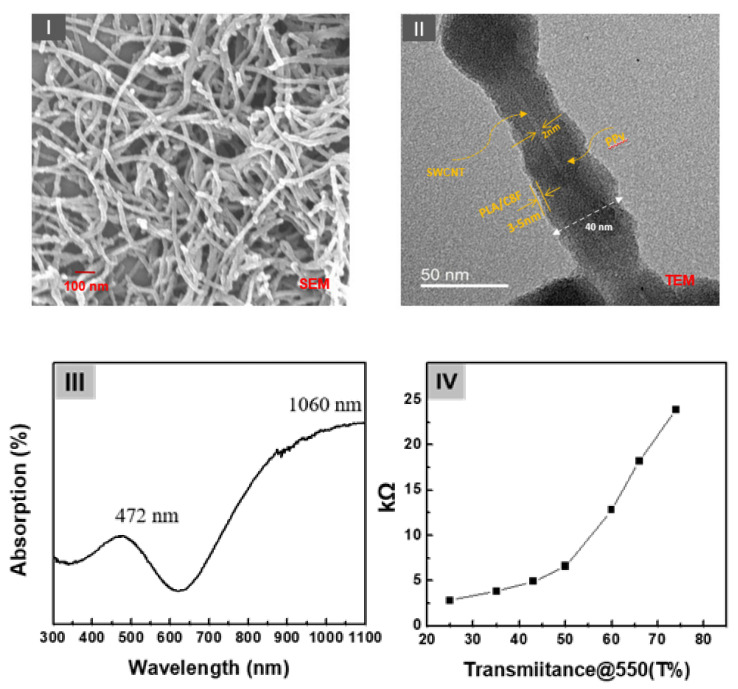
(**I**) The scanning electron microscopy (SEM) and (**II**) transmission core–shell-shaped nanorods (sample). (**III**) The surface plasmon absorption spectrum of the samples and (**IV**) the electrical resistance (Ω) of the sensor sample of the as-cast film formed as a function of the relative thickness (thickness converted concerning the light transmittance (%) at a 550 nm wavelength of the casting film).

**Figure 3 biosensors-12-00354-f003:**
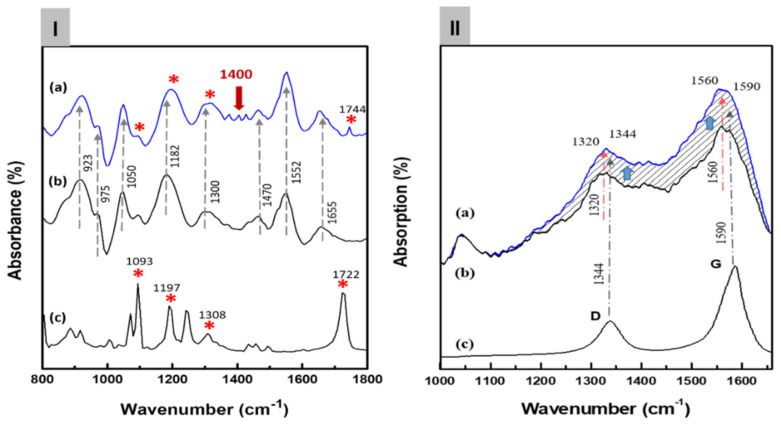
(**I**) The Fourier transform infrared (FT-IR) spectra of the C8F-doped-PPy/PLA@ SWCNT nanocomposite: (**a**) the C8F-doped-PPy/PLA@ SWCNT composite, (**b**) the C8F-doped PPy@SWCNT nanorods, and (**c**) the PLA. In the figure, the meaning of ‘*’ indicates the IR absorp-tion peak of PLA molecules. The presence or absence of PLA present in the surface layer of C8F-doped-PPy/PLA@ SWCNT nanocomposite is expressed. (**II**) The FT-Raman spectra of the C8F-doped-PPy/PLA@ SWCNT nanocomposite: (**a**) the C8F-doped-PPy/PLA@ SWCNT composite, (**b**) the C8F-doped-PPy nanorods, and (**c**) the SWCNT. (The dotted line indicates that the Raman intensity changed as the stretching mode of C8F-doped-PPy and the tangential mode of SWCNT overlapped).

**Figure 4 biosensors-12-00354-f004:**
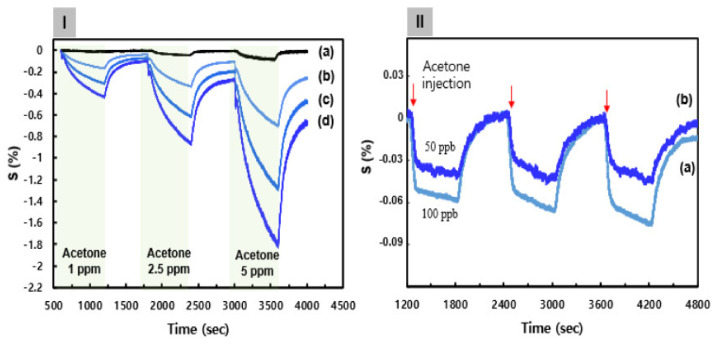
(**I**) The continuous dynamic response of the C8F-doped-PPy/PLA@SWCNT sensor to different concentrations of C_3_H_6_O (1–5 ppm) at 25 °C and in 0% RH (dry air). (**a**) The C8F-doped-PPy@ SWCNT (PLA-free). (**b**–**d**) The PLA groups were linked in a 0.1, 0.3, and 0.5 mol ratio to the positively charged PPy backbone on a C8F-doped-PPy/PLA@SWCNT nanocomposite, respectively. (**II**) The response characteristics of the C8F-doped-PPy/PLA_0.5_@SWCNT sensor with C_3_H_6_O in the 50–100 ppb concentration range at 25 °C and in 0% RH (dry air): (**a**) 100 ppb and (**b**) 50 ppb.

**Figure 5 biosensors-12-00354-f005:**
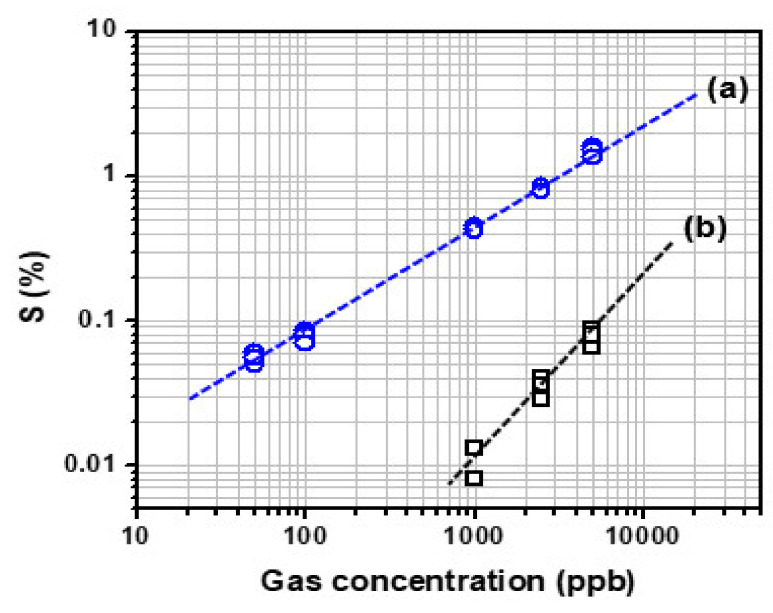
The sensitivity of the (**a**) C8F-doped-PPy/PLA_0.5_@SWCNT sensor and (**b**) C8F-doped-PPy/PLA_0.__0_@SWCNT sensor observed by exposure to 50–5000 ppb of C_3_H_6_O gas.

**Figure 6 biosensors-12-00354-f006:**
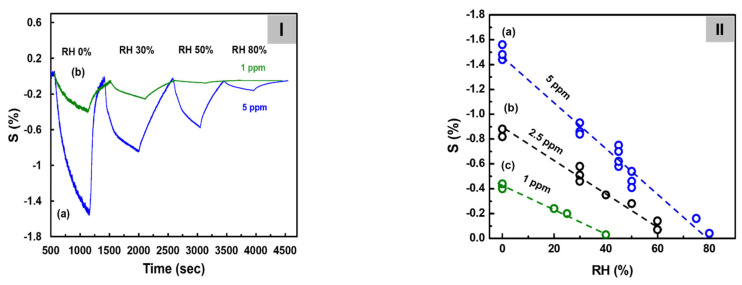
(**I**) The continuous dynamic response of the C8F_0.1_-doped-PPy/PLA_0.5_@SWCNT sensor to different humidity conditions with C_3_H_6_O gas at 5 ppm at 25 °C. (**II**) Sensitivity change with increasing humidity measured at each gas concentration: (**a**) 5 ppm, (**b**) 2.5 ppm, and (**c**) 1 ppm.

## Data Availability

Not applicable.

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
