# Peer review of "Acetone Gas Sensor Based on SWCNT/Polypyrrole/Phenyllactic Acid Nanocomposite with High Sensitivity and Humidity Stability"

_biosensors, 2022, doi:10.3390/bios12050354_

Round 1

Reviewer 1 Report

Please consider the file in attachment.

Author Response

The content of the manuscript has been corrected for the content pointed out by the reviewer.

Reviewer 2 Report

This manuscript entitled “Single-walled carbon nanotube/polypyrrole/phenyllactic acid  core-shell nanorods for high responsiveness acetone gas sensing” by Jun-Ho Byeon et al., have synthesized core–shell-shaped nanorods comprising a single-walled carbon nanotube (SWCNT) core and heptadecafluorooctanesulfonic acid-doped polypyrrole (C8F-doped-PPy)/phenyllatic acid (PLA) shells, i.e., SWCNT@C8F-doped-PPy/PLA, for detecting acetone gas with high responsiveness and moisture resistance. Such study must helpful for researchers for medical diagnoses via human breath analysis. It can be publishable on Biosensors after a minor revision after addressing following important points:

  1. Authors claimed to form nanorod morphology, however the SEM shows cylindrical tube morphology. Kindly check.
  2. In the present core-shell morphology, C8F-doped-PPy/PLA is the shell and SWCNT is core, so the denotation should be C8F-doped-PPy/PLA@SWCNT.
  3. Human body releases approx. 1 ppm of acetone and sensitivity above 40% RH is almost negligible here. So, can the author explain how such sensors can be helpful as in general RH >40% across world?
  4. Heptadecafluorooctanesulfonic acid is environmentally hazardous, how such material can be used for human medical diagnostic purpose?
  5. Authors claimed here “It was confirmed that a PPy layer of approximately 16–17 nm in thickness was surrounded at the outer surface of a single strand of a 2-nm-diameter SWCNT, and a thin organic PLA layer of 2.5 nm covered the surface layer of PPy” How PPY layer and PLA layer distinguished from the TEM image at 50 nm scale? And how the thickness determined?
  6. Authors claimed here “Using Raman spectra, we were able to prove that SWCNTs were contained in the center of the C8F-doped PPy/PLA core–shell structure, as shown in the molecular schematic in Fig. 1 and the TEM image of Fig. 2”, How Raman spectroscopy can be used to prove that SWCNT is at center? Kindly explain with supporting references.
  7. Line 210 “and the charged carriers were better activated by the expanding transport path” Kindly explain this and provide supporting data for the same.
  8. Kindly use consistent denotion for each figure, either Figure 1a, b…. or Figure 1(I), (II)….
  9. Kindly avoid using word line “ newly synthesized novel”, word responsiveness can be sensitivity, remove nanorod word etc….
  10. Typos should be corrected: Line 75”:”, line 87 “an additional” can be replaced with “for”, line 90 “:”, Some time Figure is used and some time Fig. is used, kindly correct them, Line 132 Figure 3 (left), Line 143 cm-1, Line 162 “FT-Raman, Line 184, 187 ref. number red colored, Line 280, 40 nm and 1-5 µm… References 2, 6, 13, 16, 17, 20, etc…

Author Response

(The authors gave the same response as above.)

Round 2

Reviewer 1 Report

Thank you for the corrections. I recommend publication of the manuscript in the present form.